# Red Bull Increases Heart Rate at Near Sea Level and Pulmonary Shunt Fraction at High Altitude in a Porcine Model

**DOI:** 10.3390/nu12061738

**Published:** 2020-06-10

**Authors:** Benedikt Treml, Elisabeth Schöpf, Ralf Geiger, Christian Niederwanger, Alexander Löckinger, Axel Kleinsasser, Mirjam Bachler

**Affiliations:** 1Department of General and Surgical Intensive Care, Medical University Innsbruck, 6020 Innsbruck, Austria; benedikt.treml@tirol-kliniken.at; 2Department of Anesthesiology and Critical Care Medicine, Medical University Innsbruck, 6020 Innsbruck, Austria; elisabeth.schoepf@kssg.ch (E.S.); axel.kleinsasser@tirol-kliniken.at (A.K.); 3Department for Pediatrics, Pediatrics III, Medical University Innsbruck, 6020 Innsbruck, Austria; ralf.geiger@tirol-kliniken.at; 4Department for Pediatrics, Pediatrics I, Intensive Care Unit, Medical University Innsbruck, 6020 Innsbruck, Austria; christian.niederwanger@tirol-kliniken.at; 5Hanusch Hospital, 1140 Vienna, Austria; alexander.loeckinger@oegk.at; 6Institute for Sports, Alpine Medicine and Health Tourism, Private University for Health Sciences, Medical Informatics and Technology GmbH, 6060 Hall in Tirol, Austria

**Keywords:** Red Bull energy drink, caffeine, taurine, hypoxia, ventilation/perfusion distribution, multiple inert gas elimination technique, piglets

## Abstract

Red Bull energy drink is popular among athletes, students and drivers for stimulating effects or enhancing physical performance. In previous work, Red Bull has been shown to exert manifold cardiovascular effects at rest and during exercise. Red Bull with caffeine as the main ingredient increases blood pressure in resting individuals, probably due to an increased release of (nor)-epinephrine. Red Bull has been shown to alter heart rate or leaving it unchanged. Little is known about possible effects of caffeinated energy drinks on pulmonary ventilation/perfusion distribution at sea level or at altitude. Here, we hypothesized a possible alteration of pulmonary blood flow in ambient air and in hypoxia after Red Bull consumption. We subjected eight anesthetized piglets in normoxia (FiO_2_ = 0.21) and in hypoxia (FiO_2_ = 0.13), respectively, to 10 mL/kg Red Bull ingestion. Another eight animals served as controls receiving an equivalent amount of saline. In addition to cardiovascular data, ventilation/perfusion distribution of the lung was assessed by using the multiple inert gas elimination technique (MIGET). Heart rate increased in normoxic conditions but was not different from controls in acute short-term hypoxia after oral Red Bull ingestion in piglets. For the first time, we demonstrate an increased fraction of pulmonary shunt with unchanged distribution of pulmonary blood flow after Red Bull administration in acute short-term hypoxia. In summary, these findings do not oppose moderate consumption of caffeinated energy drinks even at altitude at rest and during exercise.

## 1. Introduction

After market release of Red Bull energy drink in April 1987 in Austria, caffeinated energy drinks have become increasingly popular. During leisure time and in training, these energy drinks are popular among athletes, students and drivers for stimulating effects or enhancing physical performance.

Beyond caffeine, one of the most popular beverages worldwide, these sugary drinks are enriched with additives like taurine and glucuronolactone to a variable amount. Vitamins and minerals are contained to a smaller amount. These ingredients account for a manifold of effects [1]. In previous work, Red Bull has been shown to exert cardiovascular effects at rest and during exercise [2,3,4,5,6,7,8,9,10,11].

Plenty of data demonstrate a change of heart rate to a minor extent. Already one century ago, the earliest studies in regards to the effect of caffeine on heart rate were reviewed [12]. Two decade ago, Alford et al. demonstrated a rise of heart rate after ingestion of a 250 mL can of Red Bull [2]. This is in line with work of Grasser et al., who demonstrated an increased heart rate and cardiac output after 355 mL of Red Bull at rest in 25 young healthy adults [3]. Moreover, such an increased heart rate has been shown during a mental arithmetic task in 20 young healthy adults [7]. A current randomized, crossover trial subjecting 38 healthy adults to several caffeinated energy drinks showed an increased mean arterial blood pressure (MAP), QTc interval and heart rate [13]. These findings are supported by other studies [4,5,11]. Aside of that data two studies reported an unchanged heart rate [8,9]. A larger trial examining 68 young adults at rest demonstrated even a small drop in heart rate [10].

In regards to 24-h blood pressure, a study demonstrated an increased 24-h blood pressure in nine healthy normotensive adults after 250 mL Red Bull compared to an equivalent amount of caffeine [4]. This is in line with a larger study demonstrating an increased heart rate and systolic blood pressure (SBP) and diastolic blood pressure (DBP) in 50 subjects early after 355 mL RB ingestion [5]. Furthermore, this group could exclude any acute effects on ventricular repolarization or conventional electrocardiographic parameters, such as the PR-, QRS-, QT-, and QTc-intervals [5]. Earlier work by Ragsdale, showing an unchanged blood pressure after 250 mL Red Bull ingestion, further impedes interpretation of data [6].

In addition to the reported cardiovascular effects at rest, Alford demonstrated an improved aerobic endurance and an improved anaerobic endurance during cycling after use of Red Bull [2]. In triathletes, an increased stroke volume and left ventricular end diastolic diameter after exercise has been shown. The researchers of this small-scale study (with 13 subjects) concluded that taurine, either alone or in combination with caffeine, is responsible for this increased contractility of the left atrium [11]. More recently, Doerner and co-workers demonstrated an increased left ventricular contractility assessed by cardiac magnetic resonance in 32 healthy resting volunteers [8].

Given those cardiovascular influences after energy drink consumption, sojourning to high altitude may further amplify physiologic responses to hypoxia. At high altitude the reduced inspired paO_2_ leads to hypoxia-mediated smooth muscle contraction in the pulmonary vasculature. This hypoxic pulmonary vasoconstriction (HPV) occurs without impairment in gas exchange. If caffeinated energy drinks lead to ventilation/perfusion mismatching, it could further aggravate hypoxemia. Moreover, hypoxic induced demand tachycardia may further rise after Red Bull consumption. The current knowledge concerning effects of caffeine consumption at altitude is limited to four small studies. Those trials show an increased endurance performance in hypoxic conditions [14,15,16,17]. However, no data exist about possible effects of caffeinated energy drinks on pulmonary ventilation/perfusion distribution at altitude. The scarcity of literature encountered when designing the study led us to believe that this porcine model would be suitable for such a first pilot trial. Here, we hypothesized a possible effect of Red Bull in regards to ventilation/perfusion distribution in hypoxia. We subjected eight anesthetized piglets in normoxia (FiO_2_ = 0.21) and in hypoxia (FiO_2_ = 0.13) to 10 mL/kg Red Bull ingestion, whereas another eight served as controls. In addition to standard cardiovascular data, ventilation/perfusion distribution of the lung assessed by using the multiple inert gas elimination technique (MIGET) was measured.

## 2. Materials and Methods

### 2.1. Subjects

All experiments conformed to the guidelines of the National Institute of Health and were approved on 27.09.2006 by the Austrian Federal Animal Investigational Committee (# GZ66.011/127-BrGT/2006).

Sixteen healthy, 4-week-old cross breed piglets (German land-race x Pietrain) of either sex were selected from a local stock regularly used for experimental research. Prior to the experiment, all animals were examined in the barn by a veterinarian to affirm healthy piglets. Acute sickness of piglets would have led to exclusion. After arrival at the animal research facility, they were housed in a quiet and dark barn. Thereafter, premedication was administered intramuscularly (im.) using Ketamine (20 mg/kg) and Azaperon (4 mg/kg). Additionally, Atropine im. (0.01 mg/kg) was used to oppose a possible ketamine-induced hypersalivation. After venepuncture of an ear, anaesthesia was inducted with Propofol (4 mg/kg iv.) followed by tracheal intubation. Anaesthesia was maintained using Propofol iv. (1–1.5 mg/kg/h; Perfusor, B. Braun Melsungen AG, Melsungen, Germany) and several boluses of 15 mg Piritramide in order to keep all piglets deeply anesthetized.

All piglets were mechanically ventilated in a volume-controlled mode (Evita 2, Dräger, 23,558 Lübeck, Germany) with a fraction of inspired oxygen (FiO_2_) of 0.21 and a positive end-expiratory pressure (PEEP) set at 5 mmHg. Tidal volumes (TV) were adjusted to values of 10 mL/kg at 15 breaths/min.

Throughout the procedure, Ringer’s solution (6 mL/kg/h) and a 3% gelatine solution (4 mL/kg/h) served as compensation for any loss of blood and fluid. However, blood loss was less than 100 mL during instrumentation. A standard lead II ECG was used to monitor cardiac rhythm. Body temperature was maintained between 38 and 39 °C by using an electric heating blanket.

An 8.5 French pulmonary artery catheter was advanced from the internal jugular vein into the pulmonary artery to measure mean pulmonary arterial pressure (mPAP), pulmonary capillary wedge pressure (PCWP), central venous pressure (CVP) and cardiac output by the thermodilution technique (10 mL saline in triplicate) and to withdraw mixed venous blood. A 6.0 French arterial catheter introduced into the femoral artery was used to monitor systemic blood pressure and to take blood samples. All catheters were filled with saline and connected to pressure transducers zeroed to ambient pressure at the level of the right atrium.

### 2.2. Experimental Protocol

After surgical preparation, the animals were moved into prone position and left for a stabilization phase of 30 min. Thereafter the first baseline measurements (i.e., 0 min) including hemodynamics, blood gases, ventilation and MIGET were taken.

Sixteen animals were randomly assigned into two groups. Eight animals received Red Bull (10 mL/kg) over a stomach tube (Charriere 10, length 125 cm, Maersk Medical A/S, Roskilde, Denmark), whereas another eight were sham-tested with an equivalent volume of saline. All animals were mechanically ventilated, spontaneous breathing efforts were not observed. Thirty min after baseline, the first experimental leg started exposing all animals to ambient air (i.e., normobaric normoxia). After 120 min, the second experimental leg started with taking a second baseline. Thereafter, all animals were exposed to normobaric hypoxia with an FiO2 of 0.13 simulating an altitude of 14.000 feet (equivalent to 4.267 m). After 150 min, animals assigned to the Red Bull group received another bolus of Red Bull (5 mL/kg), whereas the remaining six received the corresponding amount of saline. Hemodynamics and blood gas samples were performed every 30 min. MIGET was performed at 0, 30, 90, 120, 150 and 240 min.

We used a standard 250 mL can of Red Bull energy drink, containing 80 mg caffeine, 1.000 mg taurine, 0.6 g glucuronolactone, 27 g carbohydrate (glucose, sucrose), 20 mg niacin (vitamin B3), 5 mg pantothenic acid (vitamin B5), 5 mg vitamin B6, 5 µg vitamin B1 and 2 µg vitamin B12, respectively [18]. Given the weight of the piglets (23.59 +/− 3.33 kg), 250 mL Red Bull resulted in a 3-fold higher dose compared to the same volume in adults.

Respirator settings were not changed during the observation period in none of the animals. After 240 min, the study ended with killing the animals using pentobarbital and potassium chloride. No adverse events in the animals were observed.

Thereafter, the heart and the proximal parts of the great arteries were removed and inspected to rule out any anatomical intra- or extracardial shunt connection.

The multiple inert gas technique was applied in the usual manner as described by [19,20,21]. In brief, six inert gases, prepared in sterile 0.9% saline, were infused continuously. After obtaining a steady state after 30 min, the six gas concentrations in expired gas, arterial and venous blood samples were assessed by gas chromatography (Hewlett-Packard 5890, series II). Using the ratios of excretion and retention allows to calculate ventilation/perfusion distribution using a least squares best fit regression analysis.

### 2.3. Statistical Analysis

A two-way ANOVA was used to determine inter- and intragroup differences. Significant results were analysed post hoc with the Newman–Keuls and Fisher’s exact tests. Results are given as mean and standard deviation, values of *p* ≤ 0.05 were considered significant.

## 3. Results

Eight anesthetized piglets were subjected to Red Bull in ambient air followed by normobaric hypoxia. Another eight animals were sham-tested with an equivalent amount of saline in normoxia followed by hypoxia.

Baseline data showed no differences between the two groups.

### 3.1. Hemodynamic Changes

Red Bull administration increased heart rate after 30 min in normoxia. In contrast, heart rate remained unchanged in normoxic controls. After induction of hypoxia, heart rate increased in the Red Bull group but remained unchanged after the second ingestion of Red Bull. In hypoxic controls, heart rate showed a somewhat delayed course with a lesser increase after induction of hypoxia compared to the verum group. Moreover, placebo treatment showed a trend towards heart rate increase reaching comparable values of the Red Bull group until the end of hypoxia (Figure 1).

MPAP increased nearly 1.5-fold after start of hypoxia in both groups with a further increase in the Red Bull group after the second dose of Red Bull at 210 and 240 min (in intragroup comparison). Placebo treatment did not change mPAP in controls. Cardiac output (CO) remained unchanged during normoxia in both groups. 30 min after induction of hypoxia CO increased in the Red Bull group only. Thereafter, these verum animals showed lower values until the end of the experiment. CO in controls showed a trend towards higher values after 210 min compared to the Red Bull group, but this failed to reach significance (Table 1).

### 3.2. Bloodgas Changes

Arterial partial pressure of oxygen (paO_2_) and mixed venous partial pressure of oxygen (pvO_2_) remained unchanged after oral Red Bull ingestion in normoxia. Shortly after exposure to hypoxia arterial partial pressure of oxygen (paO_2_) and mixed venous partial pressure of oxygen (pvO_2_) decreased in both groups. The second administration of Red Bull or placebo did not change paO_2_. However, PvO_2_ was smaller at the end of the experiment after 240 min in the Red Bull group.

The first administration of Red Bull increased arterial partial pressure of carbon dioxide (paCO_2_) after 30 min in normoxia. During the rest of the experiment, paCO_2_ in animals subjected to Red Bull was higher than in the placebo group without reaching significance. In the Red Bull group, mixed venous partial pressure of carbon dioxide (pvCO_2_) was greater after 30 min in normoxia compared to controls and baseline values. Thereafter, pvCO_2_ in the verum group was greater after 90 min in normoxia and after 210 and 240 min in hypoxia, respectively (Table 2).

### 3.3. Ventilation/Perfusion Changes

In normoxic conditions, ingestion of Red Bull or placebo did not change ventilation/perfusion distribution. Furthermore, induction of hypoxia did not change these variables. After the second Red Bull dose, blood flow to unventilated lung units (i.e., the fraction of shunt) increased after 240 min (Figure 2).

Moreover, in these verum animals, blood flow to normally ventilated units (normal V_A_/Q of Q) showed a trend towards decrease after 240 min without reaching significance. The mean of the distribution of perfusion (mean of Q) showed a trend towards a small increase, but this also did not reach significance. The distribution of pulmonary blood flow (expressed as the logarithmic deviation of the standard of the mean of the distribution of perfusion, i.e., LogSDQ) remained unchanged (Table 3).

In the MIGET, adequacy of fit of the data to the model is assessed by the remaining sum of squares (RSS). RSS was ≥ 10.6 in 91.2% of all MIGET analysis, thereby indicating good data quality.

## 4. Discussion

The salient finding of this study is that, in a controlled laboratory trial, Red Bull significantly increased pulmonary shunt fraction with unchanged distribution of pulmonary blood flow after Red Bull consumption during short-term exposure to acute hypoxia. Moreover, we could demonstrate an increased heart rate after high dose Red Bull ingestion at near sea level in piglets.

The increased heart rate after Red Bull ingestion is in line with previous work performed in man [2,3,4,5,7,11,13,22]. We observed a larger increase of heart rate as we chose a threefold higher amount of Red Bull compared to studies conducted in adults. In contrast to those studies, two recent trials reported an unchanged heart rate [8,9]. In total, 67 adults received Red Bull doses of 168 mL/m^2^ body surface area (BSA) corresponding to a volume of about 350 mL.

Ten years ago, Ragsdale and co-workers reported a trend towards drop of heart rate after 250 mL Red Bull ingestion in 68 undergraduate students [6].

Recent work examining 44 young Iranians at rest demonstrated a small drop in heart rate after Red Bull administration [10]. However, Hajsadeghi et al. used 250 mL but did not report the weight of the subjects. A study conducted over two decades ago reported a drop in heart rate in 10 endurance athletes after consumption of half a litre of Red Bull during submaximal exercise [23]. Bichler et al. observed a decreased heart rate shortly after administration of pills containing caffeine and taurine in college-students [24].

In summary, a comparison of trials reporting changes in heart rate still remains difficult given the different approaches used, variable volumes of Red Bull or any potential influences of sex category or age. Moreover, the extent of common caffeine consumption plays a pivotal role. As early as 1981, frequent caffeine consumers have been shown to respond less to acute administration than caffeine-naive individuals [25]. However, since piglets are life-long caffeine abstinent, we did not take regular caffeine consumption into account.

Acute hypoxia induces demand tachycardia as one mechanism of short-term adaption in preserving convective oxygen transport [26]. Here, we observed an accelerated heart rate at once in acute hypoxic exposure in the verum group only. Moreover, cardiac output at this time point was also higher, most likely due to the tachycardic response and a preserved stroke volume [27]. One could guess that consumption of Red Bull at high altitude may further accelerate tachycardia. Thus, maximum heart rate could be reached earlier being detrimental for oxygen delivery at high altitude. However, after the second dose of Red Bull, heart rate showed a trend to rise without any significance. Moreover, it remained nearly unchanged thereafter. In contrast, placebo treated piglets showed a smaller increase of heart rate at commencement of hypoxia. Until the end of the experiment, this group reached nearly the same heart rate as the verum group. In summary, our findings demonstrate that a high dose of Red Bull does not worsen tachycardia at high altitude.

Staying at high altitude exposes to reduced inspired paO_2_ without impairment in gas exchange. Whether energy drink consumption at high altitude impairs pulmonary blood flow and thus may limit hypoxic exercise performance is still not answered. Up to date, only four small studies reporting caffeine consumption in hypoxic conditions exist, all of them reporting an increased endurance performance [14,15,16,17]. Nearly twenty years ago, researchers Berglund and Hemmingsson reported an improved exercise performance at 2900 m. They subjected fourteen well-trained cross-country skiers to 6 mg/kg caffeine during competition in a time trial over 21 km [14]. Later on, Fulco and co-workers used a smaller caffeine dose (4 mg/kg) in eight adults cycling at 4300 m. These researchers demonstrated a prolonged time to exhaustion at 80% of the altitude-specific maximal oxygen consumption (VO_2max_) [15]. A small controlled study investigated the ergogenic effect of ingestion of 4 mg/kg caffeine in seven male adults in moderate hypoxia. At simulated 2500 m, caffeine prolonged time to exhaustion and increased the heart rate to a greater extent during high-intensity cycling than in controls. Moreover, the authors excluded a caffeine-associated reduction in neuromuscular fatigue during performance at moderate altitude [16]. Stadheim and colleagues demonstrated a prolonged time to exhaustion during double poling at sub-maximal exercise in 2000 m. However, the 13 cross-country skiers consumed half as much caffeine (4.5 mg/kg) as used in our study [17].

Those four studies used caffeine only and focused on exercise performance. In the present study, we were interested to know if Red Bull, a caffeinated energy drink, impacts pulmonary blood flow or HPV. We observed an increased shunt fraction after Red Bull consumption during acute exposure to short-term hypoxia. Furthermore, we observed increased pvCO_2_ values shortly after Red Bull ingestion in hypoxia. A possible explanation could be a decreased pulmonary CO_2_ elimination due to shunt which can be excluded here as arterial gas exchange appeared to be unchanged. The 20-fold higher diffusion capacity of CO_2_ compared to O_2_ makes this even more unlikely. In addition to that, the increased shunt did not correspond with a decreased normal V_A_/Q (only a trend) as one could expect. Moreover, we did not observe a further worsening of gas exchange or ventilation/perfusion distribution in hypoxia. In summary, Red Bull consumption did not alter gas exchange at high altitude in piglets. However, careful transfer of our findings to humans is coercible.

The respiratory stimulant nature of caffeine is long known. Before World War I, Cushney showed an increased respiratory rate to carbon dioxide after caffeine [28]. These data have been confirmed some decades later [29,30]. Moderate caffeine doses (around 3 mg/kg) have been demonstrated to increase alveolar ventilation while exercising moderately [31]. Another group showed a rise of hypoxic ventilatory response (HVR) using high doses (nearly 10 mg/kg) [32].

Recently, Cavka and co-workers were able to demonstrate an increased respiratory rate and flow rate, respectively, at rest and during moderate exercise after ingestion of half a litre of Red Bull in 38 college students. These researchers showed an activation of the sympathetic nervous system and hypothesized a notable dilation of bronchioles [33]. Recent data from Spanish colleagues showed an increased cycling performance in ambient air as well as enhanced muscle oxygen saturation assessed at the thigh at moderate workloads (30–60% of VO_2max_) after caffeine intake (3 mg/kg). Moreover, peak pulmonary ventilation during exercise and blood lactate after exercise were higher [34]. The most likely candidate for these effects is caffeine without any certain evidence up to date. Here, we did not observe an influence of Red Bull on respiratory drive as paCO_2_ and ventilation of the anesthetized piglets remained unchanged until termination of the experiment.

From a physiological point of view the intake of an ample amount of sugar (here around 40 g of sugars in about 375 mL Red Bull per piglet) has to be taken into account when interpreting our results. In the present study, paCO_2_ was higher after Red Bull ingestion (i.e., after 27 g of sugars) in ambient air. Such a glucose load may have led to an increased carbon dioxide production (VCO_2_) and consequently, given our fixed minute ventilation, an increased PaCO_2_ with a successive depression of pH. Moreover, it is known that combined consumption of caffeine and glucose causes a state of hyperinsulinemia and hyperlipidemia [35].

Beside the actions of caffeine and glucose, taurine needs to be considered when interpreting our results. Hypotensive effects were demonstrated in several studies using different hypertensive animal models (for reviews see [36,37]). Nearly 40 years ago, Bousquet et al. showed a drop in heart rate and blood pressure after taurine injection directly into a ventricle of the brain in cats [38]. Data in man showed a modulation in myocardial myofibrillar proteins with an inotropic effect in the failing heart secondary to cardiomyopathy in seventeen patients [39]. Recently, the first randomized, double-blind, placebo-controlled clinical trial showed a blood pressure drop after a twelve-week lasting oral taurine supplementation in prehypertensive adults. In regards to exercise performance, taurine has been shown to increase left atrial contractility after exercise in endurance-trained subjects [11]. This was demonstrated in comparison of Red Bull and a drink containing only caffeine. Those aforementioned beneficial effects of taurine may counterbalance the cardiovascular effects of the remaining ingredients of Red Bull. We can only hypothesize that taurine attenuated a possible caffeine induced blood pressure rise.

### Limits

A few limits have to be taken into account in interpretation of our results. First, the small sample size is clearly a limitation. However, we sought to reduce the number of animals needed as far as possible without curtailing statistical power. Clearly, using a murine model would allow for a greater sample size and deeper statistical analyses. Here we used this porcine model to obtain values resembling those in men (e.g., pulmonary blood pressures) and for giving us the possibility to perform the MIGET method. Second, this data recorded in piglets has to be transferred cautiously into any recommendations regarding humane consumption of caffeinated energy drinks. Third, we did not obtain blood glucose levels, which hampers interpretation of a possible glucose effect in Red Bull.

Lastly, further research, even in a murine or a humane setting, needs to differentiate between effects of caffeine and taurine alone, respectively, either at sea level or at high altitude.

## 5. Conclusions

In summary, high dose Red Bull consumption raised heart rate at near sea level in piglets. During acute exposure to short-term hypoxia, it did not worsen tachycardia. In regards to pulmonary blood flow, we demonstrated an increased pulmonary shunt fraction with unchanged distribution of pulmonary blood flow. We conclude that Red Bull did not alter gas exchange at high altitude in a porcine model of acute short-term hypoxia.

## Figures and Tables

**Figure 1 nutrients-12-01738-f001:**
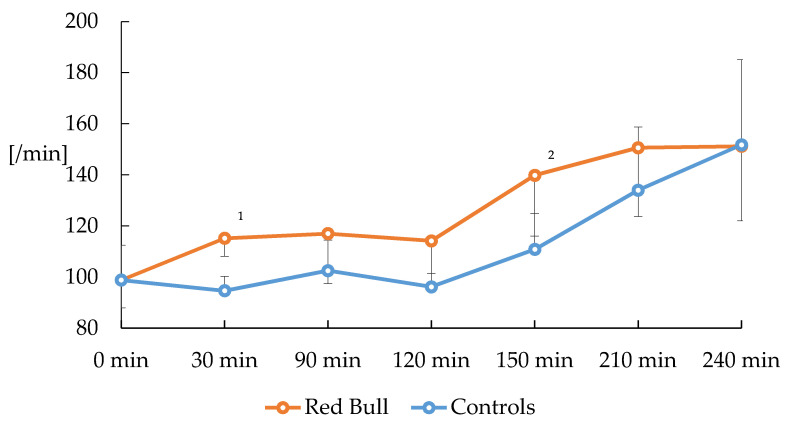
Course of heart rate after consumption of Red Bull or placebo in normoxic and hypoxic piglets (*n* = 16). ^1^
*p* < 0.01 versus control, ^2^
*p* < 0.05 versus control. Normoxia (F_i_O_2_ = 0.21) 0–120 min, Hypoxia (F_i_O_2_ = 0.13) 120–240 min. Piglets (*n* = 8) received 10 mL/kg Red Bull at 0 min and 5 mL/kg at 150 min, respectively. Values are mean ± standard deviation.

**Figure 2 nutrients-12-01738-f002:**
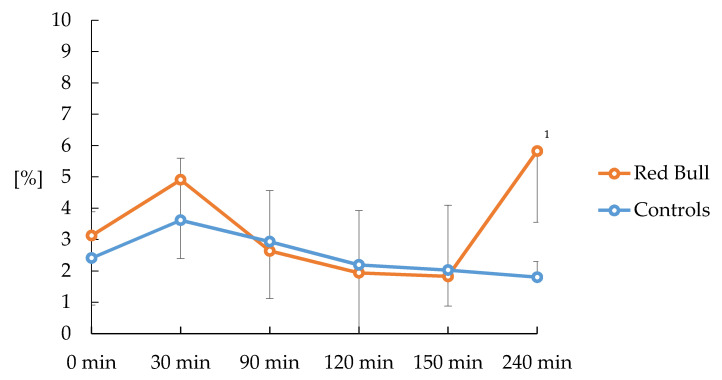
Course of shunt fraction after consumption of Red Bull or placebo in normoxic and hypoxic piglets (*n* = 16). ^1^
*p* < 0.05 versus control. Shunt reflects unventilated lung units. Normoxia (F_i_O_2_ = 0.21) 0–120 min; Hypoxia (F_i_O_2_ = 0.13) 120–240 min. Piglets (*n* = 8) received 10 mL/kg Red Bull at 0 min and 5 mL/kg at 150 min, respectively. Values are mean ± standard deviation.

**Table 1 nutrients-12-01738-t001:** Hemodynamic measurements (*n* = 16).

Time Point		0 min	30 min	90 min	120 min	150 min	210 min	240 min
		Normoxia (FiO_2_ = 0.21)			Hypoxia (FiO_2_ = 0.13)			
MAP (torr)	RB	87 ± 14	92 ± 12	97 ± 16	89 ± 12	86 ± 15	77 ± 9	82 ± 10
	control	89 ± 9	101 ± 17	98 ± 6	94 ± 12	83 ± 8	83 ± 9	86 ± 16
mPAP (torr)	RB	23 ± 3	24 ± 3	24 ± 2	24 ± 3	37 ± 4	40 ± 2 ^1^	41 ± 3 ^1^
	control	23 ± 1	24 ± 2	24 ± 2	24 ± 2	38 ± 1	38 ± 5	39 ± 4
CO (l/min)	RB	4.5 ± 0.7	4.3 ± 0.5	4.0 ± 1.0	4.2 ± 0.7	5.4 ± 0.6 ^2^	4.8 ± 0.7	4.5 ± 1.1
	control	4.5 ± 0.7	4.0 ± 0.6	4.0 ± 0.7	4.2 ± 1.2	4.3 ± 0.8	5.2 ± 1.3	4.6 ± 2.1
PCWP (torr)	RB	13 ± 0	13 ± 1	12 ± 1	10 ± 2	9 ± 3	9 ± 3	10 ± 4
	control	15 ± 2	15 ± 3	14 ± 2	12 ± 1	11 ± 1	10 ± 2	10 ± 2
CVD (torr)	RB	9 ± 1	10 ± 2	10 ± 1	9 ± 3	9 ± 3	10 ± 1	10 ± 2
	Control	10 ± 1	9 ± 1	9 ± 1	10 ± 1	11 ± 1	10 ± 2	10 ± 1

^1^*p* < 0.05 versus 150 min, ^2^
*p* < 0.05 versus control. MAP reflects mean arterial blood pressure; mPAP reflects mean pulmonary arterial blood pressure; CO reflects cardiac output; PCWP reflects pulmonary capillary wedge pressure; CVD reflects central venous pressure. RB reflects Red Bull energy drink. Piglets (*n* = 8) received 10 mL/kg Red Bull at 0 min and 5 mL/kg at 150 min, respectively. Values are mean ± standard deviation.

**Table 2 nutrients-12-01738-t002:** Blood gas measurements (*n* = 16).

Time Point		0 min	30 min	90 min	120 min	150 min	210 min	240 min
		Normoxia (FiO_2_ = 0.21)			Hypoxia (FiO_2_ = 0.13)			
paO_2_ (torr)	RB	98 ± 5	88 ± 4	88 ± 6	93 ± 4	35 ± 4	33 ± 2	33 ± 2
	control	98 ± 6	90 ± 5	87 ± 3	93 ± 3	35 ± 5	33 ± 3	34 ± 4
paCO_2_ (torr)	RB	37 ± 2	40 ± 3 ^1^	39 ± 2	37 ± 1	38 ± 4	39 ± 5	39 ± 5
	control	35 ± 2	35 ± 3	35 ± 3	36 ± 2	34 ± 5	36 ± 3	37 ± 3
pH	RB	7.44 ± 0.03	7.40 ± 0.01 ^1^	7.40 ± 0.02	7.43 ± 0.03	7.44 ± 0.04	7.33 ± 0.09	7.29 ± 0.10
	control	7.46 ± 0.02	7.46 ± 0.03	7.43 ± 0.06	7.51 ± 0.02	7.48 ± 0.04	7.43 ± 0.04	7.39 ± 0.05
pvO_2_ (torr)	RB	37 ± 2	35 ± 2	34 ± 2	34 ± 3	22 ± 1	17 ± 2	14 ± 2 ^1^
	control	36 ± 0	33 ± 2	33 ± 2	35 ± 3	22 ± 3	19 ± 2	21 ± 4
pvCO_2_ (torr)	RB	41 ± 2	46 ± 3 ^1,2^	46 ± 2 ^1,2^	43 ± 2	43 ± 4	47 ± 6 ^3^	48 ± 6 ^3^
	control	41 ± 3	40 ± 5	41 ± 4	41 ± 4	39 ± 4	40 ± 3	42 ± 3

^1^*p* < 0.05 versus control, ^2^
*p* < 0.05 versus 0 min, ^3^
*p* < 0.05 versus 150 min. PaO_2_ reflects arterial partial pressure of oxygen; paCO_2_ reflects arterial partial pressure of carbon dioxide; pH reflects arterial pH; pvO_2_ reflects mixed venous partial pressure of oxygen; pvCO_2_ reflects mixed venous partial pressure of carbon dioxide. RB reflects Red Bull energy drink. Piglets (*n* = 8) received 10 mL/kg Red Bull at 0 min and 5 mL/kg at 150 min, respectively. Values are mean ± standard deviation.

**Table 3 nutrients-12-01738-t003:** Inert gas data (*n* = 16).

Time Point		0 min	30 min	90 min	120 min	150 min	240 min
		Normoxia (FiO_2_ = 0.21)				Hypoxia (FiO_2_ = 0.13)	
Low V_A_/Q of Q (%)	RB	0 ± 0	0 ± 0	0 ± 0	0 ± 0	0 ± 0	0 ± 0
	control	0 ± 0	0 ± 0	0 ± 0	0 ± 0	1.2 ± 1.8	0.7 ± 1.1
Norm V_A_/Q of Q (%)	RB	96.8 ± 2.2	95.0 ± 2.5	97.3 ± 1.5	98.0 ± 2.5	98.1 ± 0.9	93.3 ± 2.2
	control	99.1 ± 1.4	99.2 ± 1.9	98.8 ± 1.5	97.8 ± 1.7	96.7 ± 1.7	97.0 ± 1.7
High V_A_/Q of Q (%)	RB	0 ± 0	0 ± 0	0 ± 0	0 ± 0	0 ± 0	0.5 ± 0.9
	control	0 ± 0	0 ± 0	0 ± 0	0 ± 0	0 ± 0	0 ± 0
Mean of Q	RB	0.62 ± 0.18	0.60 ± 0.16	0.62 ± 0.11	0.73 ± 0.26	0.75 ± 0.32	1.26 ± 0.54
	control	0.86 ± 0.19	0.88 ± 0.31	0.86 ± 0.20	0.63 ± 0.14	0.59 ± 0.21	0.67 ± 0.29
LogSDQ	RB	0.39 ± 0.08	0.45 ± 0.16	0.43 ± 0.06	0.36 ± 0.05	0.43 ± 0.15	0.68 ± 0.27
	control	0.45 ± 0.14	0.52 ± 0.12	0.47 ± 0.12	0.63 ± 0.20	0.57 ± 0.13	0.54 ± 0.06

Low V_A_/Q reflects lung units with a low V_A_/Q ratio; norm V_A_/Q reflects normal V_A_/Q lung units; high V_A_/Q reflects high V_A_/Q lung units; mean of Q reflects mean of the distribution of perfusion; logSDQ reflects logarithmic deviation of standard of the mean of the distribution of perfusion; RB reflects red Bull energy drink. Piglets (*n* = 8) received 10 mL/kg Red Bull at 0 min and 5 mL/kg at 150 min, respectively. Values are mean ± standard deviation.

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
