# Peer review of "Red Bull Increases Heart Rate at Near Sea Level and Pulmonary Shunt Fraction at High Altitude in a Porcine Model"

_nutrients, 2020, doi:10.3390/nu12061738_

Round 1

Reviewer 1 Report

Dear Authors,

generaly quite interesting and novelity subject. All the allegations I made while reading the manuscript the Authors carefully noted in section 4.1 Limits. Researching definitely needs to be continued. The question is why not on mice or rats? The sample could be more numerous, and the consent of the bioethical commission faster. I will recommend continuation of research also on mathematical models with greater involvement of statistical calculations.

Author Response

The reviewer adds an important point and is completely right. We have chosen a porcine rather than a murine model,
a) to obtain values resembling those in men,
b) for giving us the possibility to perform the MIGET method,
c) and due to profound expertise of our group in regards to porcine models for pulmonary hypertension or acute lung injury. This issue is now included in greater detail to the limits section (see page 9, line 330ff).

Moreover, we would like to thank the reviewer for this excellent suggestion.

Reviewer 2 Report

Overall, this is a clear, concise, and well-written manuscript. The introduction is relevant and theory based. Sufficient information about previous study findings is presented for readers to follow this study's rationale. The methods are generally appropriate. Overall, the results are clear and compelling, as noted limitations do exist to this study and interested to see further studies between the effects of caffeine and taurine alone at high altitude vs sea level.

Author Response

We would like to thank this reviewer for evaluating our manuscript.